# DDP: Dual-Decoupled Prompting for Multi-Label Class-Incremental Learning

## Abstract

Prompt-based methods have shown strong effectiveness in single-label class-incremental learning, but their direct extension to multi-label class-incremental learning (MLCIL) performs poorly due to two intrinsic challenges: semantic confusion from co-occurring categories and true-negative–false-positive confusion caused by partial labeling. We propose Dual-Decoupled Prompting (DDP), a replay-free and parameter-efficient framework that explicitly addresses both issues. DDP assigns class-specific positive–negative prompts to disentangle semantics and introduces Progressive Confidence Decoupling (PCD), a curriculum-inspired decoupling strategy that suppresses false positives. Past prompts are frozen as knowledge anchors, and interlayer prompting enhances efficiency. On MS-COCO and PASCAL VOC, DDP consistently outperforms prior methods and is the first replay-free MLCIL approach to exceed 80% mAP and 70% F1 under the standard MS-COCO B40-C10 benchmark. Our code will be open-sourced.

## 1 Introduction

Multi-label class-incremental learning (MLCIL) (Dong et al., 2023; Du et al., 2024b) is a challenging paradigm in which a model must learn to recognize multiple co-occurring classes while the label space continuously expands over time. Unlike single-label class-incremental learning (SLCIL) (Buzzega et al., 2020; Smith et al., 2023; Huang et al., 2025), each image in MLCIL may contain multiple classes, better reflecting real-world complexity but also increasing the difficulty of the learning task (Hu et al., 2023). MLCIL adopts a task-level partial labeling scheme (Du et al., 2025; Zhang et al., 2025), where only the current task labels are provided, while annotations for previous and future tasks are unavailable. This scheme stems from the nature of class-incremental learning (CIL), where knowledge from the past and future is unavailable during training.

Over the past few years, **prompt-based CIL** has emerged as a replay-free and parameter-efficient strategy for adapting powerful pre-trained encoders while mitigating catastrophic forgetting (Wang et al., 2022b;a; Smith et al., 2023; De Min et al., 2024; Huang et al., 2025). In SLCIL, methods such as L2P (Wang et al., 2022b) and DualPrompt (Wang et al., 2022a) deploy selected prompts for feature encoding, resulting in a **many-to-many** association between prompts and classes, i.e., a prompt may guide multiple classes, and a class may be guided by multiple prompts (see Figure 1 (a)). *While prompting has gained traction in SLCIL, its application to MLCIL remains nascent.* For MLCIL, MULTI-LANE (De Min et al., 2024) (see Figure 1 (b)) constructs task-specific routes that implement a **one-to-many** mapping in which a single prompt within a task is shared by all classes. Such *class-agnostic designs*, whether many-to-many or one-to-many, inevitably blur class-specific cues in multi-label settings, making them prone to confusion among co-occurring categories.

The weakness of class-agnostic prompting in MLCIL can be understood more concretely through two inherent challenges. *First*, they cannot separate fine-grained semantics among co-occurring categories. For example, in Figure 1 (b), the same one-to-many prompt (De Min et al., 2024) is shared by "person" and "dog". When recognizing "person", features related to "dog" are also activated, blurring distinctions and inducing **semantic confusion** (Zhang et al., 2025). *Second*, under task-level partial labeling, many negative classes remain unannotated, yet class-agnostic prompts lack mechanisms to calibrate confidence (Du et al., 2024b; 2025). As shown in Figure 1 (a–b), the model may assign high confidence to "car" even when it is absent, producing false positives (FP) and failing to preserve true negatives (TN). This **TN–FP confusion** inflates the false-positive rate (FPR)

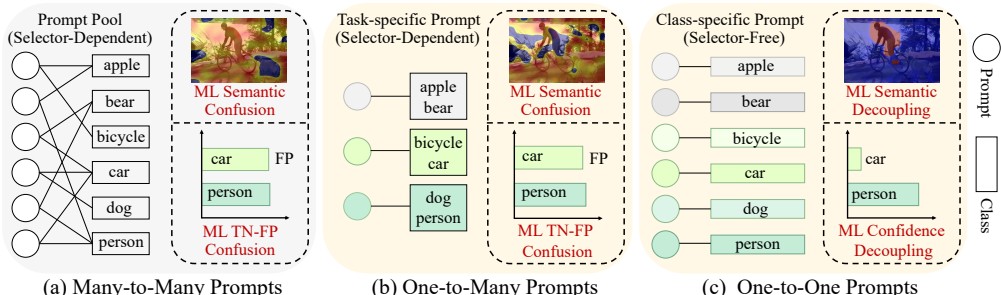

Figure 1: Comparison of prompt-based methods: two multi-label (ML) confusions and decoupling.

and severely degrades performance. Moreover, these class-agnostic approaches require selectors or auxiliary training classifiers, which increase computational cost (Huang et al., 2024). Together, these issues make class-agnostic prompting fundamentally ill-suited for MLCIL, underscoring the need for a class-specific and confidence-aware prompting framework.

To decouple semantic confusion and TN–FP confusion, we propose a selector-free Dual-Decoupled Prompting (DDP) framework for prompt-based MLCIL as shown in Figure 1 (c). First, DDP performs semantic decoupling via **one-to-one** class-specific bimodal prompting. This suppresses semantic confusion among co-occurring categories by isolating per-class semantics, yielding class-dependent signals. Second, DDP performs confidence decoupling by applying a Progressive Confidence Decoupling (PCD) strategy that dynamically adjusts confidence in a curriculum-inspired (Bengio et al., 2009) manner, effectively mitigating TN–FP confusion. Specifically tailored for prompt-based methods in MLCIL, PCD enhances both average and last performances. For efficiency, we adopt interlayer prompting, optimizing both the position and manner of prompt attachment within the visual encoder to achieve substantial spatio-temporal gains, without training selectors and classification heads in (De Min et al., 2024). In addition, to address the general challenge of catastrophic forgetting, DDP preserves the class-specific prompts learned from previous tasks as knowledge-preserving prompts, which retain highly discriminative knowledge to mitigate forgetting without replay. Our contributions are summarized in three-fold:

(1) We propose the Dual-Decoupled Prompting (DDP) framework for replay-free MLCIL, which explicitly addresses the intrinsic dual-form confusion beyond forgetting by introducing semantic and confidence decoupling within a unified prompting paradigm.

(2) DDP introduces class-specific prompting in both text and visual modalities for semantic decoupling and a PCD strategy for confidence decoupling that suppresses false positives at inference. It adopts interlayer prompting to improve efficiency.

(3) Extensive experiments on MS-COCO and PASCAL VOC benchmarks demonstrate that DDP consistently outperforms recent state-of-the-arts, and is the first replay-free MLCIL method to surpass 80% mAP and 70% F1 scores in the widely adopted MS-COCO B40-C10 setting.

## 2 RELATED WORK

**Single-label class-incremental learning (SLCIL).** Recent years have witnessed substantial progress in SLCIL. Regularization-based approaches (Kirkpatrick et al., 2017; Li & Hoiem, 2017; Zhou et al., 2022; Zhao et al., 2023; Mohamed et al., 2023) alleviate forgetting by adding penalty terms to the loss. EWC (Kirkpatrick et al., 2017) constrains important parameters identified by the Fisher information matrix, while oEWC (Schwarz et al., 2018) refines the estimation of parameter importance. Replay-based methods (Rebuffi et al., 2017; Cha et al., 2023; Rolnick et al., 2019) mitigate forgetting by maintaining a exemplar memory and mixing past samples with those from the current task during training. ER (Rolnick et al., 2019) simply rehearses stored exemplars, whereas iCaRL (Rebuffi et al., 2017) adopts a herding strategy to select representative exemplars. In addition, prompt-based methods leverage pre-trained models to enhance performance. L2P (Wang et al., 2022b) and DualPrompt (Wang et al., 2022a) learn a prompt pool and select them for different inputs through a many-to-many mechanism. CODA-P (Smith et al., 2023) further decomposes prompts into smaller learnable units that are optimized in an end-to-end manner. MG-CLIP (Huang et al., 2025) is the state-of-the-art SLCIL method, which mitigates the modality gap in CLIP-based CIL by introducing a compensation mechanism.

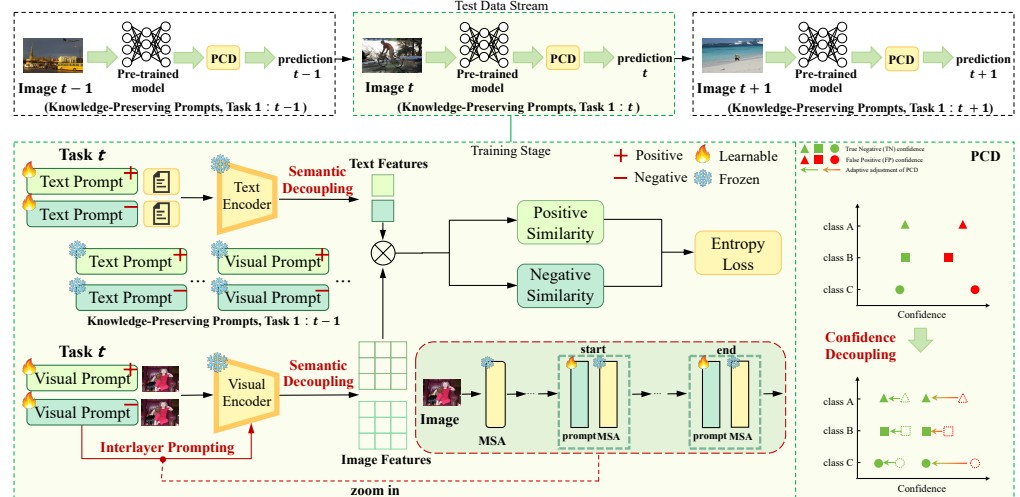

Figure 2: The overall pipeline of DDP. Training stage (bottom): prompts from past tasks are frozen as knowledge-preserving prompts, while class-specific positive–negative prompts are optimized to achieve semantic decoupling. Interlayer prompting attaches these prompts only to the last few layers for efficiency. Inference stage (top): all learned prompts are employed with PCD. PCD (right): false positives confused with true negatives are progressively suppressed and separated, reducing FPR.

**Multi-label class-incremental learning (MLCIL).** MLCIL has recently gained increasing attention. Replay-based approaches such as PRS (Kim et al., 2020) and OCDM (Liang & Li, 2022) adopt tailored sampling strategies to alleviate the challenges of long-tailed label distributions, while CUTER (Wang et al., 2025) introduces a cut-out and experience replay strategy. Regularization-based methods, AGCN (Du et al., 2024a) exploits pseudo-labels to construct statistical label correlations within a GCN, whereas KRT (Dong et al., 2023) proposes a knowledge restoration and transfer framework using cross-attention. CSC (Du et al., 2024b) employs a learnable GCN to recalibrate label dependencies and incorporates entropy regularization. HCP (Zhang et al., 2025) reduces ambiguity between known and unknown knowledge. RebLL (Du et al., 2025) tackles the intrinsic positive–negative imbalance via asymmetric loss. Meanwhile, prompt-based methods have also emerged as promising solutions. MULTI-LANE (De Min et al., 2024) introduces patch selectors and one-to-many task-specific prompts, whereas DPA (Zhao et al., 2024) leverages visual–language models with the experience replay to alleviate forgetting.

## 3 METHOD

### 3.1 PROBLEM FORMULATION AND OVERALL FRAMEWORK

**Problem formulation.** MLCIL is defined over a sequence of $T$ tasks. Each task $t \in \{1, \ldots, T\}$ is associated with a training set $\mathcal{D}^t_{\text{trn}}$ and a testing set $\mathcal{D}^t_{\text{tst}}$, with its own label space $\mathcal{C}^t$. The cumulative label space up to task $t$ is denoted as $\mathcal{C}^{1:t} = \bigcup_{i=1}^{t} \mathcal{C}^i$, where the label sets are disjoint, $\mathcal{C}^i \cap \mathcal{C}^j = \varnothing$ for $i \neq j$. Under the task-level partial labeling, only the current label set $\mathcal{Y}^t_{\text{trn}} = \mathcal{C}^t$ is available during training. At inference step $t$, the model is required to predict over the expanded label space $\mathcal{Y}^t_{\text{tst}} = \mathcal{C}^{1:t}$, ensuring that after completing task $t$ it can recognize all categories encountered so far.

**Overall framework.** Building on recent CLIP-based CIL advances (Zhao et al., 2024; Huang et al., 2025), we ground our method in a pre-trained CLIP model (Radford et al., 2021) with frozen visual encoder $E_V$ and text encoder $E_T$. Prior prompt-based methods typically realize many-to-many (Wang et al., 2022b;a) or one-to-many (De Min et al., 2024) design, failing to address inherent semantic confusion and TN–FP confusion. Figure 2 overviews our Dual-Decoupled Prompting (DDP). DDP adopts a one-to-one correspondence between prompts and classes and addresses MLCIL through two complementary components, semantic decoupling and confidence decoupling, implemented end-to-end via prompting. The details are as follows.

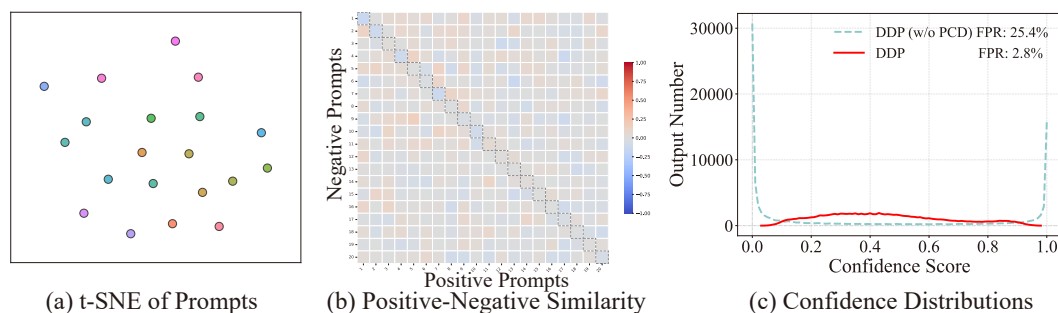

(a) t-SNE of Prompts     (b) Positive-Negative Similarity     (c) Confidence Distributions

Figure 3: Evidence for Decoupling in DDP. (a) t-SNE of class-specific prompts, showing clearer separation of fine-grained semantics. (b) Heatmap of similarities between positive-negative prompts. (c) Confidence distributions with and without PCD.

## 3.2 DUAL-DECOUPLED PROMPTING

### 3.2.1 POSITIVE-NEGATIVE CLASS-SPECIFIC PROMPTING FOR SEMANTIC DECOUPLING

Task-specific prompts (De Min et al., 2024) adopt a one-to-many design, where the same parameters are shared across categories. This class-agnostic strategy fails to disentangle co-occurring features and thus cannot resolve semantic confusion when multiple categories *overlap* within an image (Zhang et al., 2025). To explicitly achieve semantic decoupling, we propose *class-specific positive–negative prompts* for both the text and vision encoders (Figure 2). For each category $c$, we construct a one-to-one prompt set $\mathcal{P}^c = \{\mathbf{P}_{\mathrm{T}}^c, \mathbf{P}_{\mathrm{V}}^c\}$, where $\mathbf{P}_{\mathrm{T}}^c = \{\mathbf{P}_{\mathrm{T}}^{c+}, \mathbf{P}_{\mathrm{T}}^{c-}\}$ are text prompts and $\mathbf{P}_{\mathrm{V}}^c = \{\mathbf{P}_{\mathrm{V}}^{c+}, \mathbf{P}_{\mathrm{V}}^{c-}\}$ are visual prompts. Here, the "+" prompt encodes the presence of class $c$ and boosts its similarity, whereas the "−" prompt encodes the absence of $c$. This dual design reformulates multi-label recognition as a set of binary classification tasks, ensuring that each category is modeled with both inclusion and exclusion cues. The effectiveness of this design is illustrated in Figure 3: (a) t-SNE visualization shows that positive prompts $\mathbf{P}_{\mathrm{V}}^{c+}$ form well-separated clusters across 20 categories, validating the one-to-one mapping; (b) the similarity heatmap confirms that each positive prompt is clearly distinguished from its negative counterpart, as indicated by low diagonal values. Together, these results demonstrate that class-specific positive–negative prompts achieve effective semantic decoupling.

To align textual and visual features for binary classification, we compute two similarities as illustrated in Figure 2: a positive similarity $s^{c+}$ that measures evidence for the presence of $c$, and a negative similarity $s^{c-}$ that measures evidence for its absence. For each category $c \in \mathcal{C}^t$ and image $x$, class-specific text features are obtained by combining the category name with the positive and negative text prompts in $\mathrm{E}_{\mathrm{T}}$, while class-specific visual features are obtained by injecting the corresponding prompts into $\mathrm{E}_{\mathrm{V}}$. The two modalities are then aligned through cosine similarity:

$$s^{c+} = \cos(\mathrm{E}_{\mathrm{T}}(\mathbf{P}_{\mathrm{T}}^{c+}, c), \mathrm{E}_{\mathrm{V}}(\mathbf{P}_{\mathrm{V}}^{c+}, x)), \quad s^{c-} = \cos(\mathrm{E}_{\mathrm{T}}(\mathbf{P}_{\mathrm{T}}^{c-}, c), \mathrm{E}_{\mathrm{V}}(\mathbf{P}_{\mathrm{V}}^{c-}, x)), \quad c \in \mathcal{C}^t. \quad (1)$$

The pair of similarities $(s^{c+}, s^{c-})$ is normalized by a confidence-adjusted binary softmax:

$$\hat{y}_{c+}^t = \frac{\exp(s^{c+}/\tau(t))}{\exp(s^{c+}/\tau(t)) + \exp(s^{c-}/\tau(t))}, \quad c \in \mathcal{C}^t. \quad (2)$$

Here, $\hat{y}_{c+}^t$ denotes the confidence that class $c$ is present in image $x$, while the negative prediction is defined as $\hat{y}_{c-}^t = 1 - \hat{y}_{c+}^t$. The factor $\tau(t)$ acts as a task-dependent progressive scaler that increases as tasks proceed. This scaling mechanism gradually adjusts prediction confidence and forms the basis of *confidence decoupling*, which will be described in the following section.

### 3.2.2 PCD FOR CONFIDENCE DECOUPLING

While semantic decoupling clarifies *what* to attend to, it does not determine *how certain* the model should be. In MLCIL, task-level partial labeling inevitably induces confusion between true negatives (TN) and false positives (FP), which manifests as a high false-positive rate (FPR) (Du et al., 2024b; 2025). This effect can be observed in the polarized confidence distribution of Figure 3

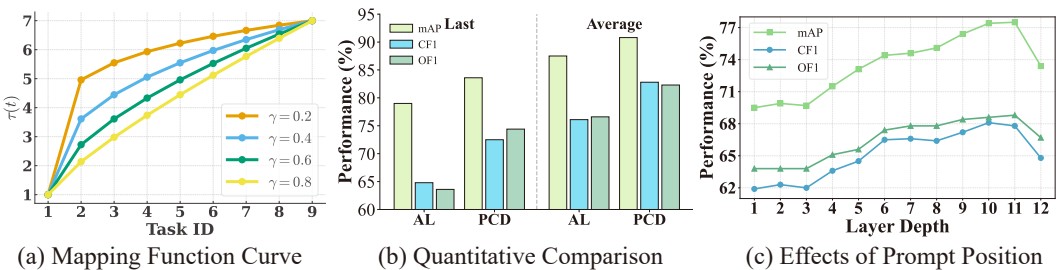

(a) Mapping Function Curve  (b) Quantitative Comparison  (c) Effects of Prompt Position

Figure 4: (a) Illustration of the PCD mapping functions under different values of $\gamma$ (with $\tau_{\max}$ set to 7), showing how prediction confidence is progressively modulated as the class number increases. (b) Comparison between AL and PCD. (c) Effects of positions for attaching prompts in the MS-COCO B0-C10 scenario, showing that deeper layers are more suitable for class-specific prompts.

(c), where DDP without confidence decoupling yields an FPR of 25.4%. Existing approaches for FPR suppression, such as asymmetric loss (AL) (Du et al., 2025), adjusts learning objectives by down-weighting positives. However, in prompt-based MLCIL where encoders are frozen and only lightweight prompts are trainable, weakening positive learning risks undermining representation quality and class-specific separability, limiting the suitability of loss-based solutions such as AL.

We therefore propose Progressive Confidence Decoupling (**PCD**), a strategy tailored for prompt-based MLCIL. PCD leaves representation learning intact and instead decouples prediction confidence at inference. Concretely, it applies the progressive scaling factor $\tau(t)$ in the binary softmax of Eq. 2, gradually adjusting the balance between positive and negative confidence. The schedule of $\tau(t)$ follows a curriculum design. As tasks accumulate, unobserved negatives become more prevalent, and a stronger adjustment is required to suppress false positives. $\tau(t)$ is defined as:

$$\tau(t) = 1 + (\tau_{\max} - 1) \cdot \left( \frac{|\mathcal{C}^{1:t}| - |\mathcal{C}^1|}{|\mathcal{C}^{1:T}| - |\mathcal{C}^1|} \right)^{\gamma}, \quad \text{s.t. } \tau_{\max} > 1, 0 < \gamma < 1, \tag{3}$$

where $|\mathcal{C}^{1:t}|$ is the cumulative number of observed classes up to task $t$ and $|\mathcal{C}^{1:T}|$ is the total number of classes. Here, $\tau_{\max}$ sets the ceiling of confidence adjustment for the final-task model, while $\gamma$ controls the schedule of adjustment over tasks: smaller $\gamma$ (closer to 0) accelerates early growth of $\tau(t)$ and yields stronger early FPR suppression, whereas larger $\gamma$ (closer to 1) produces a more gradual adjustment across the sequence. Figure 4 (a) shows the mapping function curve of $\tau(t)$. As tasks progress, $\tau(t)$ increases from 1 to $\tau_{\max}$, implementing the curriculum-inspired progression of confidence decoupling. The proof of PCD can be found in Appendix F. As shown in Figure 4 (b), PCD outperforms AL in the VOC B4-C2 scenario. Figure 3 (c) DDP (w/ PCD) shows the confidence distribution after confidence decoupling, where the FPR is reduced to 2.8%.

In summary, DDP assigns each class a pair of positive and negative prompts to align text–vision similarities, ensuring that the model *sees correctly*. PCD further decouples prediction confidence as tasks accumulate, enabling the model to *decide correctly*. Together, these two components close the loop from semantic understanding to confidence judgment.

### 3.3 EFFICIENT OPTIMIZATION

Inspired by prior work (Smith et al., 2023; De Min et al., 2024) that improves efficiency by attaching prompts to only a subset of layers, we investigate this strategy in class-specific DDP and introduce interlayer prompting shown in Figure 2, specifically:

(1) **Attachment position of prompts**: we carefully investigate different insertion positions and observe in Figure 4 (c) that deeper layers deliver stronger performance. In contrast to prior class-agnostic methods (Smith et al., 2023; De Min et al., 2024) that place prompts in shallow layers for efficiency, deeper layers provide richer semantic information and are therefore better suited to class-specific DDP. Based on this exploration, we adopt the last five layers for prompt insertion, which reduces the number of backpropagated layers and thereby improves training temporal efficiency.

(2) **Attachment manner of prompts**: for each category $c \in \mathcal{C}^t$, we concatenate the class-specific prompt embeddings $\mathbf{P}^c \in \mathbb{R}^{L_P \times d}$ (omitting positive/negative subscripts for brevity) with the hid-

den states $\boldsymbol{h} \in \mathbb{R}^{L \times d}$, where $L_P$ is the sequence length of prompts and $d$ denotes the embedding dimension. In this way, the prompts are attached to the multi-head self-attention (MSA) layer:

$$f(\mathbf{P}^c, \boldsymbol{h}) = \text{MSA}\big([\mathbf{P}^c \,; \boldsymbol{h}_Q], \, [\mathbf{P}^c \,; \boldsymbol{h}_K], \, [\mathbf{P}^c \,; \boldsymbol{h}_V]\big). \tag{4}$$

Here, $\boldsymbol{h}_Q$, $\boldsymbol{h}_K$ and $\boldsymbol{h}_V$ denote the query, key, and value for the MSA. We slice the output as $f(\mathbf{P}^c, \boldsymbol{h})[L_P :] \in \mathbb{R}^{L \times d}$ to keep dimensional consistency, enabling subsequent layers to match the backbone while still leveraging prompt interactions. Instead of assigning separate prompts to each layer (Wang et al., 2022a; Smith et al., 2023), we use interlayer shared prompts, which improves spatial efficiency. More details can be found in Appendix C. Moreover, prior many-to-many or one-to-many methods depend on the selector, increasing computational cost (Huang et al., 2024), in contrast, our DDP framework is selector-free.

Finally, we adopt a binary cross-entropy (BCE) objective to optimize DDP:

$$L_{\text{bce}} = \sum_{c \in \mathcal{C}^t} -y_c^t \log(\hat{y}_{c+}^t) - (1 - y_c^t) \log(\hat{y}_{c-}^t), \tag{5}$$

where $y_c^t \in \{0, 1\}$ denotes the ground-truth of category $c$ at task $t$. The terms $\hat{y}_{c+}^t$ and $\hat{y}_{c-}^t$ represent the predicted probabilities obtained from the positive and negative similarity scores. This objective enables parameter-efficient and end-to-end training.

## 4 EXPERIMENT

### 4.1 EXPERIMENTAL SETTINGS

**Dataset and Evaluation.** Following the experimental settings in (Dong et al., 2023; Du et al., 2024b), we evaluate our approach on MS-COCO 2014 (Lin et al., 2014) and PASCAL VOC 2007 (Everingham et al., 2010). MS-COCO 2014 is a standard benchmark for multi-label classification, containing 82,081 training and 40,504 validation images spanning 80 categories. VOC provides 20 categories with 5,011 training and 4,952 test images. We report mean average precision (mAP), class-wise F1 score (CF1), and overall F1 score (OF1). For each of these metrics, we provide the last model performance (Last) and the average performance (Avg.) across all incremental tasks.

**Experimental Setup.** Following prior MLCIL studies (Dong et al., 2023; Du et al., 2025; Zhang et al., 2025), we describe different MLCIL settings using the notation Bx-Cy, where "x" denotes the number of classes in the base task and "y" the number of new classes introduced per incremental task. On MS-COCO, we consider two standard scenarios (B0-C10, B40-C10) and two more challenging ones (B0-C5, B20-C4). For VOC 2007, we adopt B0-C4, B10-C2 as standard cases and B5-C3, B4-C2 as more difficult settings.

**Implementation Details.** The pre-trained CLIP model with ViT-B/16 (Radford et al., 2021) is used as our backbone. The size of input images is $224 \times 224$. We use Adam (Kingma & Ba, 2015) to optimize the network with $\beta_1 = 0.9$ and $\beta_2 = 0.999$. The length of prompts is set to 16. We train the prompts for 20 epochs. The incremental sequence follows the lexicographic order of class names, with the same threshold as in (Dong et al., 2023; Du et al., 2024b).

### 4.2 OVERALL PERFORMANCE

We compare twenty-two representative CIL methods spanning replay-, regularization-, and prompt-based families. For fairness, following KRT (Dong et al., 2023) and CSC (Du et al., 2024b), we use Fine-Tuning and Joint as the lower and upper bounds. Notably, the pool includes the CLIP-based state-of-the-art MG-CLIP (Huang et al., 2025) for SLCIL, the CLIP-based DPA (Zhao et al., 2024) and MULTI-LANE (De Min et al., 2024) for MLCIL. As well as strong non-prompt methods CSC, HCP (Zhang et al., 2025), RebLL (Du et al., 2025) and CUTER (Wang et al., 2025).

**MS-COCO.** Compared with replay-based and regularization-based approaches, the MS-COCO results under the B40-C10 and B0-C10 settings are summarized in Table 1. Our replay-free DDP is the first MLCIL method to surpass 80% in both last and average mAP, and to exceed 70% in CF1 and OF1 in the widely adopted B40-C10 scenario. In particular, relative to HCP, DDP achieves gains of **6.0%** in last mAP (75.3% → 81.3%), **6.6%** in CF1 (64.9% → 71.5%), **3.0%** in OF1 (68.6% → 71.6%), and **5.3%** in average mAP (78.9% → 84.2%).

Table 1: Multi-label class-incremental results on MS-COCO datasets (%). A memory size of 0 means replay-free. Type B, S, and M denote the method types: baseline, SLCIL, and MLCIL.

| Method | Memory | MS-COCO B40-C10 | | | | MS-COCO B0-C10 | | | | Type |
|---|---|---|---|---|---|---|---|---|---|---|
| | | Last | | | Avg. | Last | | | Avg. | |
| | | mAP | CF1 | OF1 | mAP | mAP | CF1 | OF1 | mAP | |
| Joint | 0 | 81.8 | 76.4 | 79.4 | - | 81.8 | 76.4 | 79.4 | - | B |
| Fine-Tuning | | 17.0 | 6.0 | 13.6 | 35.1 | 16.9 | 6.1 | 13.4 | 38.3 | B |
| TPCIL (Tao et al., 2020) | | 53.1 | 25.3 | 25.1 | 63.1 | 50.8 | 20.1 | 21.6 | 63.8 | S |
| PODNet (Douillard et al., 2020) | 5/class | 57.8 | 24.2 | 23.4 | 65.4 | 53.4 | 13.6 | 17.3 | 65.7 | S |
| DER++ (Buzzega et al., 2020) | | 59.0 | 41.9 | 43.7 | 69.6 | 54.6 | 33.3 | 36.7 | 68.1 | S |
| KRT-R (Dong et al., 2023) | | 74.3 | 66.0 | 65.9 | 78.0 | 68.3 | 60.0 | 61.0 | 75.8 | M |
| iCaRL (Rebuffi et al., 2017) | | 55.7 | 22.1 | 25.5 | 65.6 | 43.8 | 19.3 | 22.8 | 59.7 | S |
| BiC (Wu et al., 2019) | | 55.9 | 38.1 | 40.7 | 65.5 | 51.1 | 31.0 | 38.1 | 65.0 | S |
| ER (Rolnick et al., 2019) | 20/class | 61.6 | 58.6 | 61.1 | 68.9 | 47.2 | 40.6 | 43.6 | 60.3 | S |
| PODNet (Douillard et al., 2020) | | 64.2 | 46.6 | 42.1 | 71.0 | 58.8 | 45.2 | 48.7 | 70.0 | S |
| DER++ (Buzzega et al., 2020) | | 66.3 | 51.5 | 53.5 | 73.6 | 63.1 | 45.2 | 48.7 | 72.7 | S |
| KRT-R (Dong et al., 2023) | | 75.2 | 67.9 | 68.9 | 78.3 | 70.2 | 63.9 | 64.7 | 76.5 | M |
| PRS (Kim et al., 2020) | | 33.2 | 9.3 | 15.1 | 50.8 | 27.9 | 8.5 | 14.7 | 48.8 | M |
| OCDM (Liang & Li, 2022) | | 34.0 | 9.5 | 15.5 | 51.3 | 28.5 | 8.6 | 14.9 | 49.5 | M |
| KRT-R (Dong et al., 2023) | 1000 | 75.1 | 67.5 | 68.5 | 78.3 | 69.3 | 61.6 | 63.6 | 75.7 | M |
| CSC-R (Du et al., 2024b) | | 76.0 | 67.8 | 69.7 | 78.5 | 73.9 | 67.5 | 68.5 | 79.3 | M |
| CUTER (Wang et al., 2025) | | - | - | - | - | 47.8 | 35.9 | 39.2 | 60.1 | M |
| oEWC (Schwarz et al., 2018) | | 27.3 | 11.1 | 16.5 | 44.8 | 24.3 | 6.7 | 13.4 | 46.9 | S |
| LwF (Li & Hoiem, 2017) | | 51.7 | 47.0 | 45.7 | 64.8 | 42.4 | 45.3 | 43.7 | 61.2 | S |
| AGCN (Du et al., 2024a) | | 69.1 | 58.7 | 59.9 | 73.9 | 61.4 | 53.9 | 56.6 | 72.4 | M |
| KRT (Dong et al., 2023) | 0 | 74.0 | 64.4 | 63.4 | 77.8 | 65.9 | 55.6 | 56.5 | 74.6 | M |
| CSC (Du et al., 2024b) | | 75.0 | 65.7 | 67.0 | 78.2 | 72.8 | 64.9 | 66.8 | 78.0 | M |
| HCP (Zhang et al., 2025) | | 75.3 | 68.6 | 68.9 | 78.9 | 71.2 | 60.4 | 65.3 | 77.9 | M |
| RebLL (Du et al., 2025) | | 72.5 | 60.7 | 64.9 | 76.4 | 70.4 | 60.9 | 63.8 | 77.2 | M |
| **DDP** | 0 | **81.3** | **71.5** | **71.6** | **84.2** | **78.5** | **69.0** | **70.3** | **84.1** | M |

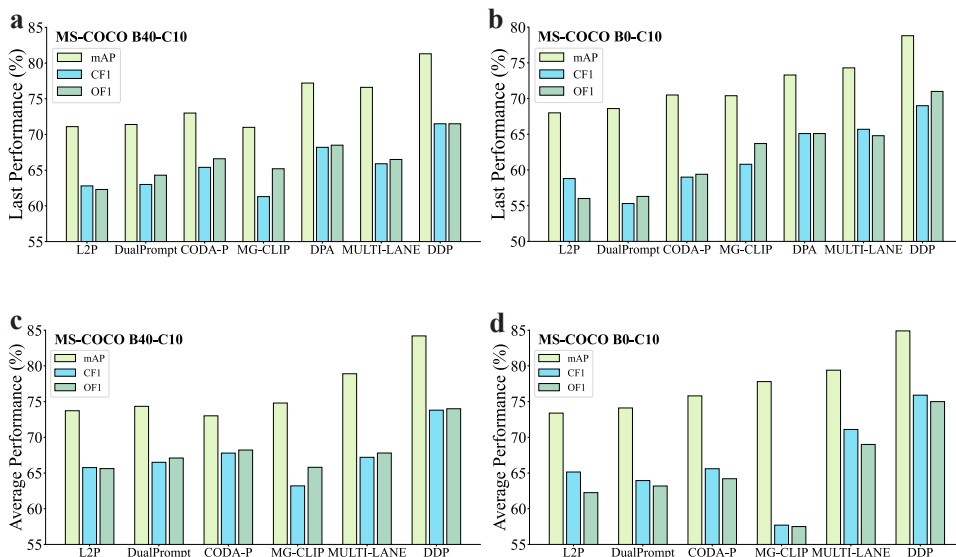

Figure 5: Comparison of prompt-based CIL methods in terms of last and average performance.

Figure 5 compares our approach with prompt-based methods, including the SLCIL methods L2P (Wang et al., 2022b), DualPrompt (Wang et al., 2022a), CODA-P (Smith et al., 2023), and MG-CLIP (Huang et al., 2025), as well as the MLCIL methods DPA (Zhao et al., 2024) and MULTI-LANE (De Min et al., 2024). Figure 5 (a) and (b) present the final performance, while (c) and (d) report the average performance. It can be observed that DDP consistently surpasses prompt-based methods across all metrics. For example, in Figure 5 (a), in the B40-C10 scenario, DDP improves the mAP from 76.6% to 81.3%, CF1 from 66.0% to 71.5%, OF1 from 66.6% to 71.6%, clearly demonstrating its advantage over MULTI-LANE.

Table 2: Multi-label class-incremental results on PASCAL VOC dataset (%).

| Method | Memory | VOC B0-C4 | | | | VOC B10-C2 | | | | Type |
| | | Last | | | Avg. | Last | | | Avg. | |
| | | mAP | CF1 | OF1 | mAP | mAP | CF1 | OF1 | mAP | |
| Joint | 0 | 93.6 | 86.0 | 88.8 | - | 93.6 | 86.0 | 88.8 | - | B |
| Fine-Tuning | | 54.0 | 37.6 | 44.2 | 80.9 | 49.4 | 30.9 | 36.9 | 74.4 | B |
| ER (Rolnick et al., 2019) | 2/class | 67.7 | 47.2 | 47.2 | 82.6 | 64.3 | 40.8 | 37.6 | 78.8 | S |
| PODNet (Douillard et al., 2020) | | 72.3 | 52.0 | 50.3 | 84.1 | 66.2 | 54.3 | 40.8 | 80.2 | S |
| DER++ (Buzzega et al., 2020) | | 74.4 | 59.8 | 56.4 | 84.9 | 67.1 | 55.5 | 45.9 | 81.6 | S |
| DPA (Zhao et al., 2024) | | 88.1 | - | - | 93.8 | 84.1 | - | - | 90.9 | M |
| CUTER (Wang et al., 2025) | 1000 | 67.9 | 51.4 | 60.0 | 82.1 | - | - | - | - | M |
| OCDM (Liang & Li, 2022) | | 45.4 | 36.4 | 40.5 | 76.1 | - | - | - | - | S |
| DualPrompt (Wang et al., 2022a) | 0 | 83.4 | 60.6 | 58.9 | 89.9 | 83.0 | 57.5 | 52.2 | 88.5 | S |
| CODA-P (Smith et al., 2023) | | 84.0 | 65.5 | 62.1 | 90.3 | 83.4 | 59.8 | 54.0 | 89.1 | S |
| KRT (Dong et al., 2023) | | 84.2 | 60.1 | 59.8 | 91.8 | 72.0 | 25.8 | 38.7 | 84.9 | M |
| MULTI-LANE (De Min et al., 2024) | | 88.8 | 74.8 | 69.9 | 93.5 | 88.0 | 63.8 | 49.4 | 93.0 | M |
| MG-CLIP (Huang et al., 2025) | | 85.4 | 72.7 | 64.4 | 92.5 | 86.4 | 52.8 | 53.5 | 90.8 | S |
| CSC (Du et al., 2024b) | | 85.1 | 67.7 | 62.2 | 90.4 | 83.8 | 62.0 | 47.2 | 89.0 | M |
| HCP (Zhang et al., 2025) | | 87.9 | - | - | 92.9 | 81.9 | - | - | 90.1 | M |
| RebLL (Du et al., 2025) | | 85.1 | 72.6 | 75.8 | 91.4 | 80.9 | 64.1 | 66.1 | 88.8 | M |
| **DDP** | 0 | **90.2** | **76.9** | **80.8** | **94.8** | **88.7** | **74.8** | **76.4** | **93.5** | M |

Table 3: Ablation of **class-specific prompts** under Last and Avg. metrics (%).

| Method | Last | | | Avg. | | |
| | mAP | CF1 | OF1 | mAP | CF1 | OF1 |
| Prompt-free | 12.0 | 11.8 | 17.7 | 14.3 | 14.4 | 18.0 |
| Global Prompt | 24.9 | 16.6 | 26.5 | 35.3 | 22.8 | 34.8 |
| Task-specific Prompt | 32.6 | 24.5 | 31.6 | 41.5 | 34.9 | 44.4 |
| **Class-specific Prompt** | 83.6 | 72.5 | 74.4 | 90.8 | 82.8 | 82.3 |

Table 4: Ablation of **positive-negative prompts** under Last and Avg. metrics (%).

| Method | Last | | | Avg. | | |
| | mAP | CF1 | OF1 | mAP | CF1 | OF1 |
| Prompt-free | 12.0 | 11.8 | 17.7 | 14.3 | 14.4 | 18.0 |
| Neg. Prompts | 76.2 | 59.3 | 58.9 | 86.0 | 70.6 | 70.1 |
| Pos. Prompts | 79.7 | 62.6 | 68.3 | 88.5 | 77.7 | 79.5 |
| **Neg. + Pos.** | 83.6 | 72.5 | 74.4 | 90.8 | 82.8 | 82.3 |

**PASCAL VOC.** Table 2 reports the results on VOC by comparing three types of methods. Our DDP consistently outperforms all baselines in both last and average performance. In the B10-C2 scenario, DDP achieves improvements of 0.7% in last mAP, 11.0% in CF1 and 27.0% in OF1 over MULTI-LANE. Furthermore, compared with the replay-based DPA with CLIP, DDP substantially outperforms it by a significant margin of 4.6% in the final mAP and 2.6% in the average mAP.

Table 5: Ablation of **PCD** under Last and Avg. metrics (%).

| Method | Last | | | Avg. | | |
| | mAP | CF1 | OF1 | mAP | CF1 | OF1 |
| Prompt-free | 12.0 | 11.8 | 17.7 | 14.3 | 14.4 | 18.0 |
| + Text Prompt | 71.1 | 40.3 | 31.7 | 81.7 | 60.2 | 50.8 |
| ++ Visual Prompt | 83.6 | 43.3 | 37.6 | 90.8 | 62.8 | 56.8 |
| **+++ PCD** | 83.6 | 72.5 | 74.4 | 90.8 | 82.8 | 82.3 |

### 4.3 Ablation Study and Analysis

**Semantic decoupling ablation.** Table 3 summarizes the ablation on different levels of semantic decoupling in the VOC B4-C2. Compared with the prompt-free baseline, introducing a global prompt (one-to-all) already improves the performance. Task-specific prompts (one-to-many) achieve higher results. Most notably, class-specific prompts (one-to-one) deliver better performance, highlighting the importance of fine-grained semantic decoupling in learning class discriminative representations.

Table 4 presents the ablation on positive-negative prompts. Using either positive or negative prompts alone provides considerable benefits compared to the prompt-free baseline. Combining both leads to better performance across all metrics.

**Confidence decoupling ablation.** Table 5 first presents the modality ablation in the VOC B4-C2, where text prompts already yield clear improvements and visual prompts bring further gains by enhancing representations. Then, we further examine confidence decoupling. With our PCD, the model suppresses over-confident predictions and reduces the FPR from 25.4% to 2.8%, while simultaneously improving both Last and Avg. performance. These results highlight the necessity of confidence decoupling for addressing the high FPR issue in MLCIL.

**Anti-forgetting performance in Multi-Task.** As shown in Figure 6, we evaluate the anti-forgetting performance of different methods in the long-sequence VOC B4-C2 and COCO B0-C5. The results

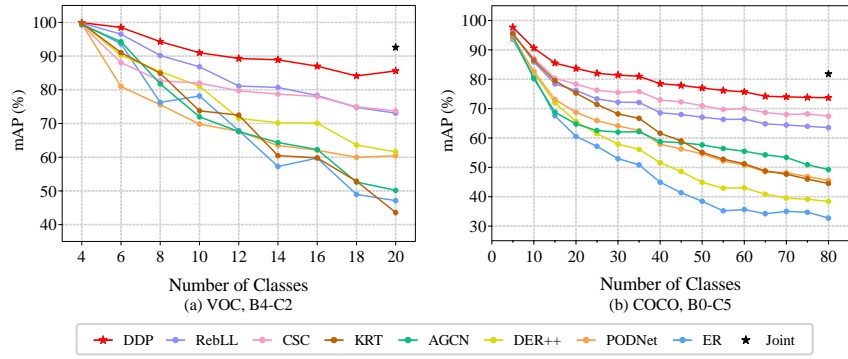

Figure 6: The results of multi-task MLCIL on challenging VOC B4-C2 and COCO B0-C5, where a larger number of tasks are involved.

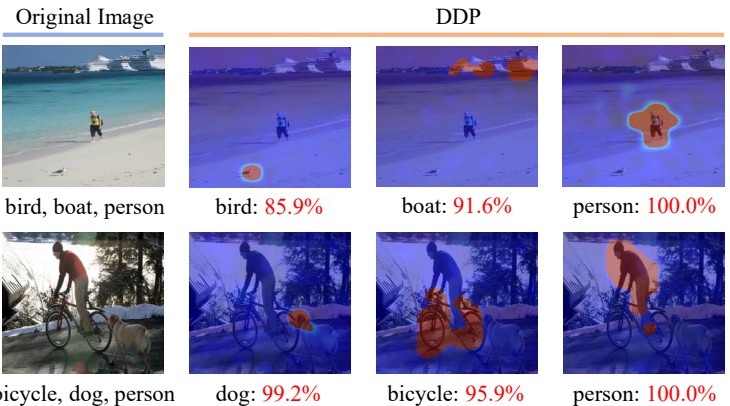

Figure 7: Visualization of DDP using the last task model.

demonstrate that our approach consistently achieves lower forgetting across tasks, indicating its stronger ability to preserve previously learned knowledge while adapting to new classes.

**Class activation maps.** As shown in Figure 7, the original images are shown in the first column, followed by class activation maps corresponding to each predicted category with confidence scores. These results are obtained by testing the final-task model on categories from different tasks, showing accurate recognition with our one-to-one prompting (see details in Appendix D).

Additional experimental results, including comparisons under more MLCIL scenarios, ablation studies, and parameter analyses, visualizations, etc., are provided in Appendix A and D.

## 5 CONCLUSION

MLCIL prompt-based methods face two intrinsic confusions: semantic confusion and TN–FP confusion. To address these issues, we propose Dual-Decoupled Prompting (DDP), a replay-free and parameter-efficient framework. DDP integrates two decoupling strategies: positive–negative class-specific prompting for semantic decoupling and PCD for confidence decoupling. In addition, inter-layer prompting optimizes both the position and the manner of prompt attachment, with DDP requiring no selector, thereby improving efficiency. Extensive experiments on MS-COCO and PASCAL VOC show that DDP consistently outperforms prior methods, and it is the first replay-free approach to exceed 80% mAP and 70% F1 scores in the widely adopted MS-COCO B40-C10 benchmark, demonstrating its effectiveness.

Limitation. We adopt vanilla BCE to optimize DDP for simplicity, leaving open the potential of tailored objectives to further enhance cross-modal alignment (see details in Appendix E).

## ETHICS STATEMENT

We affirm adherence to the ICLR Code of Ethics. This work uses only publicly available datasets, MS-COCO 2014 and PASCAL VOC 2007 for multi-label image classification, without collecting new data or involving interventions on human subjects. The experiments operate at the category level and do not attempt to identify individuals. Dataset usage and evaluation protocols follow established practice described in our experimental settings. Conflicts of interest and sponsorship: the authors declare no competing interests, any funding sources did not influence the research design, analysis, or reporting. To preserve double-blind review, specific acknowledgments will be provided.

## REPRODUCIBILITY STATEMENT

We aim to facilitate reproducibility as follows. The problem setup, model components, and training objective are described in the main text, including the binary softmax formulation for positive and negative scores (Eq. 2), the curriculum schedule of PCD (Eq. 3), and the BCE training objective (Eq. 5). The end-to-end training procedure for tasks $t = 1, \ldots, T$ is summarized in Algorithm 1. The backbone, input resolution, optimizer, prompt length, epochs, thresholding protocol, and class-incremental schedules are provided in implementation details. We report extensive ablations, PCD proof and efficiency measurements in the Appendix. To further support reproducibility, we include in the supplementary materials the full source code, configuration files, and a README, together with processing instructions for MS-COCO and PASCAL VOC.

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

Table A.1: Multi-label class-incremental results on MS-COCO dataset (%).

| Method | Memory | MS-COCO B20-C4 | | | | MS-COCO B0-C5 | | | | Type |
|---|---|---|---|---|---|---|---|---|---|---|
| | | Last | | | Avg. | Last | | | Avg. | |
| | | mAP | CF1 | OF1 | mAP | mAP | CF1 | OF1 | mAP | |
| Joint | 0 | 81.8 | 76.4 | 79.4 | - | 81.8 | 76.4 | 79.4 | - | B |
| Fine-Tuning | | 19.4 | 10.9 | 13.4 | 36.5 | 22.5 | 15.0 | 23.6 | 48.1 | B |
| ER (Rolnick et al., 2019) | | 41.9 | 32.9 | 29.8 | 53.0 | 40.1 | 32.9 | 32.3 | 54.6 | S |
| PODNet (Douillard et al., 2020) | 5/class | 58.4 | 44.0 | 39.1 | 67.7 | 58.2 | 45.1 | 40.8 | 67.2 | S |
| DER++ (Buzzega et al., 2020) | | 57.3 | 41.4 | 35.5 | 65.5 | 57.9 | 43.6 | 39.2 | 68.2 | S |
| LwF (Li & Hoiem, 2017) | | 34.6 | 17.3 | 31.8 | 55.4 | 35.9 | 30.0 | 28.0 | 54.9 | S |
| AGCN (Du et al., 2024a) | | 55.6 | 44.2 | 39.6 | 65.7 | 53.0 | 43.2 | 41.1 | 64.4 | M |
| KRT (Dong et al., 2023) | 0 | 45.2 | 17.6 | 33.0 | 64.0 | 44.5 | 22.6 | 37.5 | 63.1 | M |
| RebLL (Du et al., 2025) | | 60.1 | 51.3 | 49.2 | 69.2 | 63.5 | 53.5 | 51.9 | 71.7 | M |
| MULTI-LANE (De Min et al., 2024) | | 71.0 | 50.8 | 40.6 | 76.3 | 70.5 | 52.5 | 43.3 | 76.3 | M |
| **DDP** | 0 | **73.3** | **56.9** | **57.0** | **78.8** | **73.7** | **60.0** | **60.7** | **80.2** | M |

Table A.2: Multi-label class-incremental results on PASCAL VOC dataset (%).

| Method | Memory | VOC B4-C2 | | | | VOC B5-C3 | | | | Type |
|---|---|---|---|---|---|---|---|---|---|---|
| | | Last | | | Avg. | Last | | | Avg. | |
| | | mAP | CF1 | OF1 | mAP | mAP | CF1 | OF1 | mAP | |
| Joint | 0 | 93.6 | 86.0 | 88.8 | - | 93.6 | 86.0 | 88.8 | - | B |
| Fine-Tuning | | 37.0 | 25.0 | 27.9 | 60.4 | 49.4 | 30.9 | 36.9 | 74.4 | B |
| ER (Rolnick et al., 2019) | | 47.1 | 34.7 | 33.1 | 69.9 | 62.8 | 50.6 | 47.8 | 78.3 | S |
| PODNet (Douillard et al., 2020) | 2/class | 60.4 | 45.3 | 38.5 | 71.1 | 70.3 | 47.7 | 43.3 | 81.4 | S |
| DER++ (Buzzega et al., 2020) | | 61.6 | 33.4 | 29.9 | 77.0 | 68.1 | 53.3 | 51.4 | 78.0 | S |
| LwF (Li & Hoiem, 2017) | | 50.4 | 32.8 | 30.9 | 73.4 | 74.1 | 50.5 | 45.6 | 84.8 | S |
| AGCN (Du et al., 2024a) | | 50.2 | 35.5 | 33.5 | 71.2 | 71.6 | 55.2 | 51.1 | 83.1 | M |
| KRT (Dong et al., 2023) | 0 | 43.6 | 13.7 | 31.0 | 71.0 | 74.6 | 39.3 | 46.6 | 86.2 | M |
| RebLL (Du et al., 2025) | | 73.1 | 57.2 | 60.0 | 84.6 | 79.4 | 64.6 | 66.9 | 87.7 | M |
| MULTI-LANE (De Min et al., 2024) | | 82.1 | 43.2 | 36.1 | 89.8 | 87.1 | 66.9 | 56.8 | 92.1 | M |
| **DDP** | 0 | **83.6** | **72.5** | **74.4** | **90.8** | **88.2** | **77.1** | **80.5** | **93.0** | M |

# A ADDITIONAL EXPERIMENTAL DETAILS AND RESULTS

**The method of comparison.** We compare twenty-two CIL methods, including the replay-based methods such as iCaRL (Rebuffi et al., 2017), TPCIL (Tao et al., 2020), BiC (Wu et al., 2019), ER (Rolnick et al., 2019), PODNet (Douillard et al., 2020), DER++ (Buzzega et al., 2020), PRS (Kim et al., 2020), OCDM (Liang & Li, 2022), CUTER (Wang et al., 2025), the regularization-based methods oEWC (Schwarz et al., 2018), LwF (Li & Hoiem, 2017), AGCN (Du et al., 2024a), KRT (Dong et al., 2023), CSC (Du et al., 2024b), HCP (Zhang et al., 2025), RebLL (Du et al., 2025), and the prompt-based methods L2P (Wang et al., 2022b), DualPrompt (Wang et al., 2022a), CODA-P (Smith et al., 2023), MG-CLIP (Huang et al., 2025), MULTI-LANE (De Min et al., 2024), DPA (Zhao et al., 2024).

Note that the SLCIL state-of-the-art method MG-CLIP (Huang et al., 2025) is LoRA-CLIP–based. Its visual classifier trains on a single shared feature across classes, which causes the severe semantic confusion analyzed in our work. Even with careful learning-rate tuning, its performance remains far below our DDP shown in Figure 5 and Table 2. Therefore, these results underscore that the decoupling introduced in this paper is necessary for MLCIL.

**Main Results.** Table A.1 and Table A.2 report results under more challenging scenarios: long-sequence settings on MS-COCO (B20-C4, B0-C5) and high-frequency increments on VOC (B4-C2, B5-C3). While existing methods suffer from severe forgetting or FPR, DDP consistently achieves better last and average performance, demonstrating robust knowledge preservation and effective suppression of FPR across both datasets.

**Hyperparameter Analysis.** Our way of hyperparameter discussion and dataset usage follows the practice in the multi-label learning method (Cheng & Vasconcelos, 2024). Figure A.1 and Figure A.2 illustrate the influence of $\tau_{max}$ and $\gamma$ on PCD. On VOC B4-C2, we choose $\tau_{max}{=}7$ and $\gamma{=}0.2$,

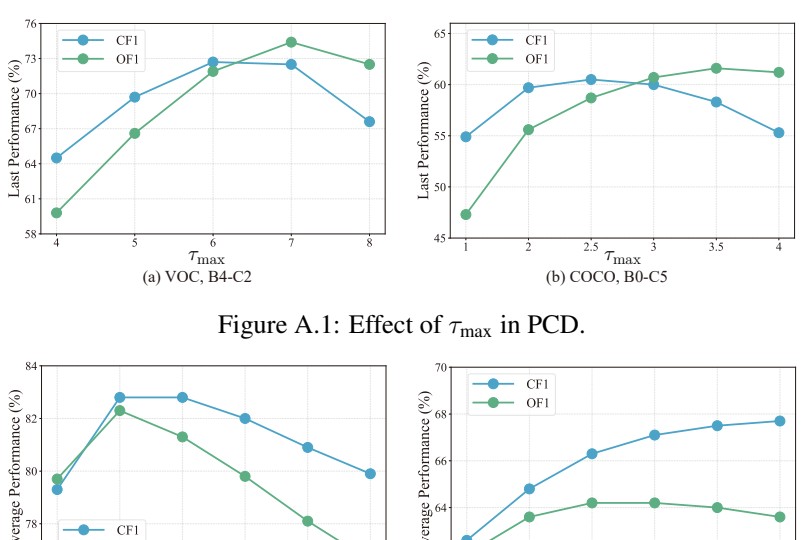

Figure A.1: Effect of $\tau_{\max}$ in PCD.

Figure A.2: Effect of $\gamma$ in PCD.

which lead to clear improvements in both Last and Avg. CF1/OF1. On COCO B0-C5, we choose $\tau_{\max}=3$ and $\gamma=0.7$, which also enhance both Last and Avg. metrics. These results indicate that PCD consistently improves performance across datasets.

**Semantic decoupling ablation.** Table A.3 reports the ablation results across different levels of semantic decoupling in the COCO B0-C5 scenario. Relative to the prompt-free baseline, the incorporation of a global prompt (one-to-all) yields a clear performance gain. Further improvements are observed with task-specific prompts (one-to-many). Better results are achieved with class-specific prompts (one-to-one), which emphasize the critical role of fine-grained semantic decoupling in enhancing discriminative representation learning. Table A.4 compares different prompt attributions under both Last and Avg. metrics in the COCO B0-C5. Employing either type individually yields substantial gains over the prompt-free baseline, with positive prompts demonstrating a more pronounced effect. Notably, the joint utilization of both results in consistently superior performance across all evaluation metrics, underscoring their complementary roles in decoupling class-level semantics and enhancing the quality of learned representations.

**Confidence decoupling ablation.** We provide results in the COCO B0-C5 in Table A.5. It shows that progressively adding text prompts, visual prompts and PCD yields consistent improvements in Last and Avg. metrics, with PCD effectively boosting performance by suppressing false positives.

**Prompt Length.** As shown in Figure A.3, we present the effect of different prompt lengths in the MS-COCO B40-C10 scenario. In this work, the text and visual prompts are assigned the same length. When $L_P = 16$, it strikes a balance between efficiency and effectiveness.

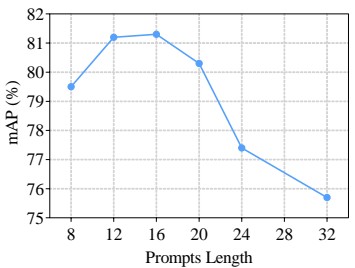

Figure A.3: Effect of prompts length.

**Forgetting Measure.** Table A.6 reports the forgetting measure in the VOC B4-C2. Our DDP shows much lower forgetting and FPR, highlighting its effectiveness in preserving past knowledge and suppressing false positives.

## B ALGORITHM

Algorithm 1 shows the training procedure that iterates over tasks $t = 1$ to $T$. For each task, DDP initializes class-specific text and vision prompt pairs $\mathbf{P}_T^c$, $\mathbf{P}_V^c$ (one pair per class). For every training

Table A.3: Ablation of **class-specific prompts** under Last and Avg. metrics (%).

| Method | Last | | | Avg. | | |
|---|---|---|---|---|---|---|
| | mAP | CF1 | OF1 | mAP | CF1 | OF1 |
| Prompt-free | 4.5 | 4.0 | 6.1 | 7.4 | 5.2 | 9.1 |
| Global prompt | 21.8 | 11.8 | 21.9 | 31.7 | 14.2 | 22.5 |
| Task-specific prompt | 36.7 | 23.7 | 39.7 | 42.1 | 26.0 | 39.7 |
| **Class-specific prompts** | 73.7 | 60.0 | 60.7 | 80.2 | 67.4 | 64.1 |

Table A.4: Ablation of **positive-negative prompts** under Last and Avg. metrics (%).

| Method | Last | | | Avg. | | |
|---|---|---|---|---|---|---|
| | mAP | CF1 | OF1 | mAP | CF1 | OF1 |
| Prompt-free | 4.5 | 4.0 | 6.1 | 7.4 | 5.2 | 9.1 |
| Neg. Prompts | 70.9 | 54.2 | 47.0 | 77.0 | 64.6 | 58.5 |
| Pos. Prompts | 71.3 | 56.2 | 51.6 | 78.5 | 66.0 | 60.2 |
| **Neg. + Pos.** | 73.7 | 60.0 | 60.7 | 80.2 | 67.4 | 64.1 |

Table A.5: Ablation of **PCD** under Last and Avg. metrics (%).

| Method | Last | | | Avg. | | |
|---|---|---|---|---|---|---|
| | mAP | CF1 | OF1 | mAP | CF1 | OF1 |
| Prompt-free | 4.5 | 4.0 | 6.1 | 7.4 | 5.2 | 9.1 |
| + Text Prompt | 65.3 | 49.8 | 43.1 | 72.4 | 60.4 | 52.7 |
| ++ Visual Prompt | 73.7 | 55.3 | 48.3 | 80.2 | 62.6 | 62.0 |
| **+++ PCD (ours)** | 73.7 | 60.0 | 60.7 | 80.2 | 67.4 | 64.1 |

Table A.6: The comparison of forgetting measure in the VOC B4-C2 (%).

| Method | mAP ↑ | Forgetting ↓ | Avg.mAP ↑ | FPR ↓ |
|---|---|---|---|---|
| LwF | 50.4 | 10.8 | 73.4 | 32.5 |
| ER | 47.1 | 11.3 | 69.9 | 33.4 |
| DDP | 83.6 | 5.4 | 90.8 | 2.8 |

sample $x^t$ and class $c \in \mathcal{C}^t$, it encodes text and image with the frozen CLIP encoders, computes cosine similarities $s^{c+}$ and $s^{c-}$, and converts them via a binary softmax to produce $\hat{y}_{c+}^t$ and $\hat{y}_{c-}^t$. We fix $\tau(t) = 1$ to ensure stability during training. A BCE objective trains only the prompts, leaving the backbone unchanged, and no prompt selector or extra classifier head is required. After optimization, the task-$t$ prompts are frozen to serve subsequent tasks.

## C  COMPARISON OF DIFFERENT ATTACHMENT POSITIONS AND ATTACHMENT MANNERS

**Comparison of different attachment positions.** We observe a clear layer position effect in Table C.1: attaching our class-specific prompts into deeper transformer layers yields a better trade-off between accuracy and compute. Since our prompts are class-specific rather than generic, they align better with the high-level semantic features emerging in deeper layers. Shallow-layer attachment (1–5) underperforms due to low-level, non-discriminative features. The 8–12 range achieves the highest mAP (78.5%) with moderate overhead, indicating that interlayer prompting benefits most from semantically rich stages. The same setting is used across other datasets and scenarios.

**Efficiency comparison**: Table C.2 reports the training cost under different backward layer settings on COCO B0-C10. Updating all layers requires noticeably higher time and memory, while our last-5 (8–12) backward strategy substantially reduces both training time and GPU consumption, highlighting the spatio-temporal efficiency of our prompt-based method.

**Comparison of different attachment methods.** Table C.3 compares the parameter efficiency, computational overhead, and performance of different prompting strategies on VOC. As shown, Prompt tuning (Wang et al., 2022b) requires back-propagation through all 12 layers, which increases the

Table C.1: Comparison of different attachment positions on COCO B0-C10. $\Delta$ FLOPs is the forward overhead vs. Prompt-free. # Params indicates the number of learnable parameters, and Backward layers refer to the transformer layers involved in back-propagation.

| Backward layer range | Backward layers | # Params (M) | $\Delta$ FLOPs (%) | mAP (%) |
|---|---|---|---|---|
| 1–5 | 5 | 1.96 | + 4.7 | 71.2 |
| 8–11 | 4 | 1.96 | + 3.7 | 76.7 |
| 9–11 | 3 | 1.96 | + 2.8 | 78.3 |
| 8–12 | 5 | 1.96 | + 4.7 | **78.5** |
| 10–12 | 3 | 1.96 | + 2.8 | 77.4 |
| 10–11 | 2 | 1.96 | + 1.9 | 77.4 |
| 1–12 (all) | 12 | 1.96 | + 11.2 | 78.0 |

---

**Algorithm 1** The training of DDP framework

---

**Require:** Training data $\mathcal{D}_{trn}^t$, Pre-trained CLIP backbone $\theta = \{E_V, E_T\}$
1: **for all** $t$ in 1 to $T$, $c \in \mathcal{C}^t$ **do**
2:     Initialize $\mathbf{P}_T^c = \{\mathbf{P}_T^{c+}, \mathbf{P}_T^{c-}\}$, $\mathbf{P}_V^c = \{\mathbf{P}_V^{c+}, \mathbf{P}_V^{c-}\}$
3:     **for all** $(x^t, y^t) \in \mathcal{D}_{trn}^t$ **do,**
4:         $s^{c+} = \cos(E_T(\mathbf{P}_T^{c+}, c), E_V(\mathbf{P}_V^{c+}, x^t))$
5:         $s^{c-} = \cos(E_T(\mathbf{P}_T^{c-}, c), E_V(\mathbf{P}_V^{c-}, x^t))$
6:         $\hat{y}_{c+}^t = \exp(s^{c+}/\tau(t))/(\exp(s^{c+}/\tau(t)) + \exp(s^{c-}/\tau(t)))$
7:         $\hat{y}_{c-}^t = \exp(s^{c-}/\tau(t))/(\exp(s^{c+}/\tau(t)) + \exp(s^{c-}/\tau(t)))$
8:         $L_{bce} = \sum -y_c^t \log(\hat{y}_{c+}^t) - (1 - y_c^t) \log(\hat{y}_{c-}^t)$
9:         Optimize $\mathbf{P}_T^c$ and $\mathbf{P}_V^c$ by $L_{bce}$
10:    **end for**
11:    Preserve $\mathbf{P}_T$, $\mathbf{P}_V$ in task $t$
12: **end for**

---

Table C.2: Comparison of the cost of different backward layers on COCO B0-C10. Time denotes the training time per batch, Memory indicates the GPU memory consumption.

| Backward layers | Time (s/batch) | Memory (GB) |
|---|---|---|
| full-12 (1-12) | 0.23 | 12.32 |
| last-5 (8-12, ours) | 0.16 | 6.95 |

training cost, while Prefix tuning (Wang et al., 2022a; Smith et al., 2023) reduces the number of back-propagated layers to 5 at the expense of introducing more parameters. In contrast, our Interlayer prompting achieves a better trade-off: it maintains the same lightweight parameter size as Prompt tuning and the reduced back-propagation depth of Prefix tuning, yet delivers the highest mAP (83.6%). These results highlight that our strategy is not only accurate but also spatio-temporally efficient, simultaneously reducing both spatial parameter cost and temporal training overhead. Table C.4 compares MULTI-LANE and our DDP in the VOC B4-C2. DDP uses fewer learnable parameters (0.49M vs. 0.80M; about 39% less) while achieving higher performance.

# D VISUALIZATION

**Class activation maps.** As shown in Figure 7, the original images are shown in the first column, followed by class activation maps (CAM) corresponding to each predicted category with confidence scores. For example, in the second row, the model correctly identifies "bicycle", "dog" and "person", and the highlighted regions precisely align with the respective objects. Notably, the body of "person", including fine-grained parts such as the legs, can be clearly delineated, demonstrating the model's ability to capture detailed semantics. In the first row, the categories "bird", "boat" and "person" are accurately detected with high confidence, and the CAM highlights the relevant regions consistently. These results are obtained by testing the final-task model on categories from different tasks, showing not only accurate recognition but also strong anti-forgetting performance.

Table C.3: Comparison of different attachment methods based on our DDP in the VOC B4-C2. $\Delta$ FLOPs is the forward overhead vs. Prompt-free. # Params indicates the number of learnable parameters, and Backward layers refer to the transformer layers involved in back-propagation.

| Method | Backward layers | # Params (M) | $\Delta$ FLOPs (%) | mAP (%) |
|---|---|---|---|---|
| Prompt-free | 0 | 0 | 0 | 12.0 |
| Prompt tuning | full-12 (1-12) | 0.49 | + 11.2 | 82.9 |
| Prefix tuning | last-5 (8-12) | 2.45 | + 4.7 | 80.4 |
| Interlayer prompting | last-5 (8-12) | 0.49 | + 4.7 | **83.6** |

Table C.4: Comparison of methods in the VOC B4-C2. # Params indicates learnable parameters.

| Method | # Params (M) | mAP (%) | CF1 (%) | OF1 (%) |
|---|---|---|---|---|
| MULTI-LANE | 0.80 | 82.1 | 43.2 | 36.1 |
| DDP (ours) | 0.49 | **83.6** | **72.5** | **74.4** |

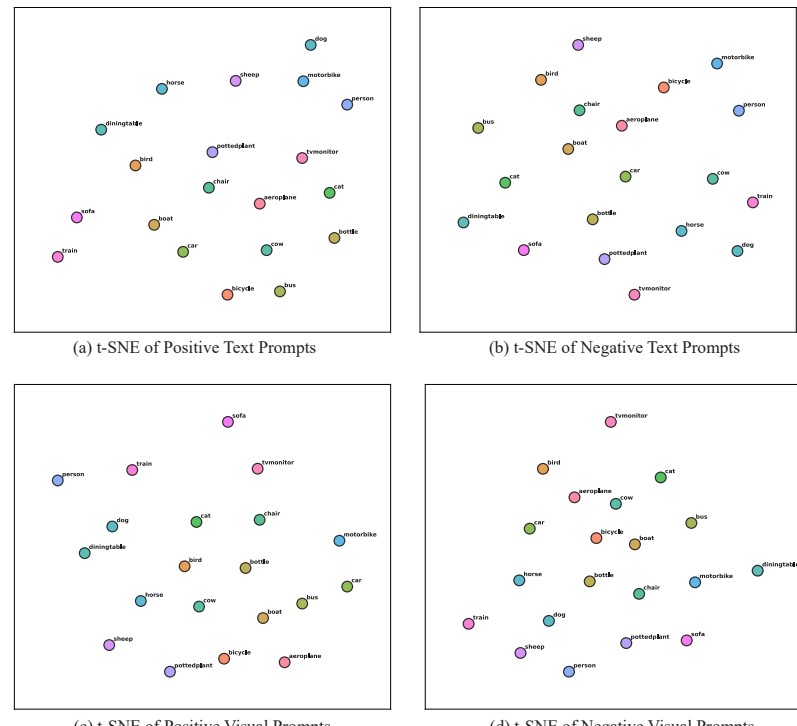

(a) t-SNE of Positive Text Prompts

(b) t-SNE of Negative Text Prompts

(c) t-SNE of Positive Visual Prompts

(d) t-SNE of Negative Visual Prompts

Figure D.1: t-SNE visualization of one-to-one class-specific prompts learned on VOC. Subplots (a) and (b) show the distributions of positive and negative text prompts, while (c) and (d) depict the corresponding visual prompts.

**t-SNE visualizations.** In Figure D.1, we present t-SNE visualizations of the learned one-to-one class-specific prompts across text and visual modalities. Positive prompts (Figures D.1 (a) and (c)) exhibit compact clusters that align well with semantic categories, suggesting their effectiveness in capturing class-presence cues. By contrast, negative prompts (Figures D.1 (b) and (d)) show a dispersed distribution, reflecting their complementary role in modeling class-absence signals. These patterns confirm that our one-to-one positive–negative prompting strategy successfully disentangles semantic information across modalities, thereby enhancing class discriminability in MLCIL.

# E  LIMITATION AND FUTURE WORK

**Limitation.** We adopt vanilla BCE to isolate the effect of our design and keep the training pipeline minimal. This choice leaves open the potential benefits of tailored objectives for catastrophic forgetting and cross-modal alignment.

**Future work.** Two future extensions are: learned confidence scheduling and loss design. For the former, replace $\tau(t)$ with a learnable scheduler $\tau_c(x, t)$ that adapts by classes or samples, and is trained under curriculum constraints to reduce false positives. For the latter, we move beyond plain BCE to systematically study objectives that improve retention and alignment.

## F  PROOF OF PCD FOR CONFIDENCE DECOUPLING

**Proposition 1** (Confidence decoupling effect of PCD). *For a class $c \in \mathcal{C}^t$ at task $t$, let the positive and negative similarity scores be $s^{c+}$ and $s^{c-}$ and define the margin $\Delta^c = s^{c+} - s^{c-}$. Progressive Confidence Decoupling (PCD) computes the positive score by a binary softmax with task-dependent scaling $\tau(t) > 1$:*

$$\hat{y}_{c+}^t(\tau) \;=\; \frac{\exp(s^{c+}/\tau)}{\exp(s^{c+}/\tau) + \exp(s^{c-}/\tau)} \;=\; \sigma\Big(\frac{\Delta^c}{\tau}\Big),$$

*where $\sigma(z) = \frac{1}{1+e^{-z}}$. Then:*

*For every $\Delta^c > 0$, $\hat{y}_{c+}^t(\tau)$ is strictly decreasing in $\tau$:*

$$\frac{\partial \, \hat{y}_{c+}^t(\tau)}{\partial \tau} = -\frac{\Delta^c}{\tau^2}\, \sigma\Big(\frac{\Delta^c}{\tau}\Big)\Big(1 - \sigma\Big(\frac{\Delta^c}{\tau}\Big)\Big) \;<\; 0.$$

*For a decision threshold $\eta$, when $\hat{y}_{c+}^t \geq \eta$, the model predicts the presence of class $c$. $\hat{y}_{c+}^t(\tau) \geq \eta$ is equivalent to a margin condition*

$$\hat{y}_{c+}^t(\tau) \geq \eta \iff \frac{\Delta^c}{\tau} \geq \sigma^{-1}(\eta) \iff \Delta^c \geq \tau \cdot s_\eta,$$

*where $s_\eta := \sigma^{-1}(\eta) = \log\big(\frac{\eta}{1-\eta}\big)$. Thus, increasing $\tau$ raises the required margin linearly.*

*In the case of a negative class ($y_c^t = 0$), a high $\hat{y}_{c+}^t \geq \eta$ leads to a false positive (FP). Consider a negative sample with margin $\Delta^c > 0$ that is a false positive at $\tau = 1$, $\sigma(\Delta^c) \geq \eta \iff \Delta^c \geq s_\eta$.*

*Then, for any $\tau' > 1$ such that*

$$\Delta^c \;<\; \tau' \, s_\eta,$$

*the prediction under PCD with $\tau \geq \tau'$ satisfies $\hat{y}_{c+}^t(\tau) = \sigma(\Delta^c/\tau) < \eta$, so the sample is corrected from a false positive (FP) to a true negative (TN), thereby reducing the overall FPR.*

As shown in Figure 3 (b), before PCD the confidence distribution on negative classes is highly polarized, with many scores near 1, causing TN–FP confusion, which means confidence entanglement. PCD introduces scaling $\tau(t) > 1$ so that $\hat{y}_{c+}^t(\tau) = \sigma(\Delta^c/\tau)$ is softened. Near the decision boundary, this softening lowers over-confident negatives below $\eta$, thereby separating TN from FP, achieving confidence decoupling.

**Curriculum schedule.**  At inference, we apply Progressive Confidence Decoupling (PCD) by using $\tau(t) > 1$ as a curriculum-inspired *post-hoc* adjustment:

$$\tau(t) \;=\; 1 + (\tau_{\max} - 1)\Big(\frac{|\mathcal{C}^{1:t}| - |\mathcal{C}^1|}{|\mathcal{C}^{1:T}| - |\mathcal{C}^1|}\Big)^\gamma, \quad \text{s.t. } \tau_{\max} > 1,\ 0 < \gamma < 1.$$

Here, $\tau_{\max}$ controls the final-task decoupling level, while smaller $\gamma$ accelerates early growth of $\tau(t)$, larger $\gamma$ yields a smoother progression. This curriculum-inspired design aligns the growth of confidence smoothing with the accumulation of missing negatives across tasks (Du et al., 2025) , producing lower FPR and improved average and final performance (see Figure 4 (b)). This re-scales confidence, suppressing over-confident false positives and improving performance without re-training the model. Both scaling and threshold tuning are post-hoc strategies. The latter modifies the decision boundary directly, the former reshapes the probability distribution itself. Our PCD extends this idea with a curriculum design that adapts naturally across all class-incremental tasks, yielding more stable improvements and making it particularly well-suited for MLCIL.

**Others.**  mAP is a ranking-based metric. Our PCD is a monotonic transform of the confidence. It compresses over-confident predictions, reduces FPR and improves the precision-recall trade-off, yielding better CF1 and OF1.

## G  THE USE OF LLMS

We only use lightweight LLMs to check grammatical errors.

