# OpenReview forum: "DDP: Dual-Decoupled Prompting for Multi-Label Class-Incremental Learning"
_ICLR.cc/2026/Conference — Submitted to ICLR 2026_

### Official Review · Reviewer_43au · 2025-10-16

**Soundness:** 3
**Presentation:** 3
**Contribution:** 3
**Rating:** 6
**Confidence:** 4

**Summary:**

This paper presents a novel approach called Dual-Decupled Prompting (DDP) for rehearsal-free multi-label class-incremental learning (MLCIL). DDP mitigates two primary issues of existing prompt-based methods, namely the inability to separate fine-grained semantics among co-occurring categories (i.e, prompts are optimized for multiple classes within a task), and the inability to properly calibrate confidence for partially annotated images (a known problem in MLCIL). To address the first problem, the authors propose to semantically decouple classes by exploiting "one-to-one" class prompts, i.e., creating a one-to-one mapping between learnable prompts and classes. Instead, the second issue is mitigated through a Progressive Confidence Decoupling (PCD), which dynamically adjusts the binary cross-entropy loss confidence at every task. Authors compare their method against 22 baselines, achieving a new state-of-the-art in established benchmarks.

**Strengths:**

1. The paper is overall well-written, and most parts are easy to understand.
2. The proposed method (DDP) is sound, well-motivated, and PCD is supported by theoretical justifications.
3. Performance is clearly superior compared to existing approaches.
4. I appreciated that the source code was attached to the submission together with a pseudocode in the Appendix.

**Weaknesses:**

**Major weaknesses**
1. The paper uses CLIP as the pre-trained backbone while the current state-of-the-art exploits ViT (IN-1k). I believe this is not completely fair. Thus, the paper would benefit if the authors demonstrated that the existing state-of-the-art is worse than DDP even when using CLIP. Alternatively, authors could have shown DDP performance using ViT as a backbone, although this would compromise the method's integrity. I understand these types of experiments take a lot of time; therefore, I do not expect authors to be able to run them within the limited rebuttal time.
2. It is not completely clear how $s^{c+}, s^{c-}$ are computed and the shape of $P^c$ (see question below).
3. Although this paper's novelty is unquestionable in my opinion, I feel the use of class-specific prompts shares some similarities with existing works in other fields [1,2].
4. PCD hyperparameters ($\gamma, \tau_\text{max}$) seem a bit too sensitive and I believe require good hyperparameter tuning, which is non-trivial in continual learning.

**Minor weaknesses**
1. Lines between 79 and 80, \citep{} -> \citet{}
2. Authors ablated the use of positive-only and negative-only prompts. It would have been interesting to also try Positive and negative on the vision encoder and a single prompt in the text encoder, and vice versa (two prompts for the text encoder and one for the vision).
3. Eq. 4 shows that prompts are prepended following prefix tuning. It is common in prefix tuning to only prepend/append prompts to the key and value matrices to avoid increasing the sequence length. Yet, authors also prepend them to the query matrix and then slice the output to remove the extra tokens. It is not wrong per se, but it seems odd as it adds useless extra computation.
4. I do not get why prompt-based approaches are separated from Table 1 and only appear in Figure 5.
5. Two references have missing venues (MLCIL is published at CoLLAs 2024, while DPA is at BICS 2024).
6. Figure 2 is a bit messy and fails at conveying the proposed methodology.


[1] Laiti, Francesco, et al. "Conditioned prompt-optimization for continual deepfake detection." ICPR, 2024. \
[2] Lee, Dongjun, et al. "Read-only prompt optimization for vision-language few-shot learning." ICCV, 2023.

**Questions:**

1. I do not really get how $s^{c+/-}$ are computed. Lines 269-276 explain that prompts are prepended to the multi-head self-attention in a prefix-tuning manner [3]. This creates the confusion, as the output of the MSA has the same sequence length as the input, which is also explicitly shown by the authors in L.275. Then, if the output of the transformer has the same length as the input, how can $cos(E_\text{T}(P_\text{T}^{c+/-}, c), E_\text{V}(P_\text{V}^{c+/-}, x))$ be computed? Or, in other terms, how can each vision class/prompt have its specific output that can then be compared with the corresponding textual positive/negative prompts? My guess is that the authors recompute the last vision transformers layers (where prompts are injected) multiple times, i.e., $2 (+/-) \times |\mathcal{C}^{1:t}|$. If that is the case, this could greatly increase the required computation, and should be compared with existing baselines. Another alternative is that authors do not slice the output of the last transformer layer, thus preserving the prompt length.
2. What is the prompt size at every MSA layer? The paper reports $L_P\times d$, yet it is common to have separate prompts for MSA inputs. Thus, I'd expect them to have size $3\times L_P\times d$. If it is not the case, I'd like to know why.

[3] Jia, Menglin, et al. "Visual prompt tuning." ECCV, 2022.

**Motivation for my score** \
I believe this is a good paper; yet some parts are unclear and require further details for me to confidently pick a side. Therefore, I decided to keep my initial score low. I am willing to increase it based on the author's rebuttal.

---

### Official Review · Reviewer_cami · 2025-10-24

**Soundness:** 3
**Presentation:** 3
**Contribution:** 2
**Rating:** 2
**Confidence:** 4

**Summary:**

This paper proposes a CLIP-based replay-free (a.k.a, exemplar-free) class-incremental learning (CIL) framework, namely DDP, to address the continual learning problem in the multi-label image classification task. DDP aims to address the semantic confusion problem and the TN–FP confusion problem in multi-label CIL (MLCIL). To address the semantic confusion problem, DDP introduces a positive soft prompt and a negative soft prompt in both text and visual modalities for each class. To address the TN–FP confusion, DDP introduces a progressive confidence decoupling (PCD) strategy by dynamically adjusting confidence in a curriculum-inspired manner. This paper claims that DDP consistently outperforms recent SOTA methods by conducting experiments on MS-COCO and PASCAL VOC.

**Strengths:**

1. This paper focuses on the MLCIL problem, which is challenging and important to real-world continual learning systems.
2. This paper addresses the semantic confusion in a simple and effective way by introducing a positive soft prompt and a negative soft prompt in both text and visual modalities for each class, achieving better performance than other prompt-based MLCIL methods.
3. The paper provides both mathematical and experimental analysis of how PCD reduces the FPR.

**Weaknesses:**

1. DDP is not practical for a real-world CIL system. The progressive scaling factor $\tau(t)$ of PCD depends on the total number of classes (i.e., $|\mathcal{C}^{1:T}|$), which is not accessible in a real-world continual learning system. A real-world continual learning system can't know how many tasks or classes to learn in the future.
2. The comparison in Section 4.2 is not fair as the involved methods use different backbone networks. For example, KRT (Dong et al., 2023) and HCP (Zhang et al., 2025) adopt TResNet-M (5.5GFLOPs, 29.4M Params.) pre-trained on ImageNet-21k, and MULTI-LANE uses a standard ViT-B/16 (49.3GFLOPs, 86M Params.) pretrained on ImageNet-1k. However, DDP uses the pre-trained CLIP model with ViT-B/16 as the backbone network, but the paper adopts the performance from other methods on other backbone networks, which is not fair. Therefore, we cannot verify the claim that DDP achieves SOTA performance.
3. The paper doesn't provide enough information on how the authors select the hyper-parameters ($\tau_{\text{max}}$ and $\gamma$). Comparing Figure A.1, A.2 with Table A.2, A.2, it seems that Figure A.1, A.2 shows the performance on testing datasets. The information provided in the paper does not eliminate my suspicion that the reported results were obtained by carefully tuning the hyper-parameters on testing datasets.
4. The paper lacks the implementation details on interlayer prompting. For example, the paper doesn't analyze the difference between interlayer prompting, prompt tuning, and prefix-tuning, and how interlayer prompting improves efficiency.

**Questions:**

My key concerns about this paper are listed in the Weaknesses section. Besides, I list some of my concerns that do not impact my rating:

1. Could you provide a figure like Figure 2 (bottom) or a pseudo-code like Algorithm 1 to illustrate how DDP inferences in a clearer way? What does the "pre-trained model" in Figure 2 (top) refer to?
2. The paper lacks details on how the evaluation metrics are calculated. For example, it is not clear whether the micro-F1 score or the macro-F1 score is used. It would be better if the authors could provide the formal definitions of the metrics used in this paper in the Appendix.
3. The paper does not report the error of the results. It would be better if the paper could report the standard error, standard deviation, or the 95% confidence interval of the results after repeating experiments with varying data order / random seed.
4. Please report the results of MULTI-LANE and MG-CLIP in Table 1.
5. DDP is designed for the positive–negative imbalance. Is this design also effective for mitigating the classification bias brought by the class-imbalanced data? Verifying DDP on extremely class-imbalanced datasets (e.g., long-tailed datasets) to evaluate DDP's performance may further enhance the contribution.

---

### Official Review · Reviewer_TbkU · 2025-10-26

**Soundness:** 2
**Presentation:** 1
**Contribution:** 2
**Rating:** 2
**Confidence:** 4

**Summary:**

This paper proposes the DDP, a method for the multi-label class-incremental learning (MLCIL). DDP specifies the problems of semantic confusion and TN-FP confusion in the MLCIL and proposes a one-to-one semantic decoupling and a progressive confidence decoupling (PCD) to address these issues. Experiments demonstrate the good performance of DDP in the continual learning of CLIP. However, there are several points of this paper are not well presented. For example, the motivation of this study is not clearly illustrated, relation between the recognized problem and continual learning is not clearly presented, setting of MLCIL and the requirement of number of categories in PCD is not practical and reasonable when considering the real-world senarios, and experiments are not sufficient. Therefore, I think this version does not meet the standard of ICLR and vote for rejection.

**Strengths:**

1. The proposed DDP transforms the multi-label CIL into binary classification, which is rare and noval in the class-incremental learning.
2. The proposed DDP demonstrate good performance in the setting of MLCIL in this paper.

**Weaknesses:**

1. The setting of MLCIL in this paper seems not pratical in the real-world senario. The setting obeys the task-level partial labeling, which masks the  unavaible labels in a sample. However, this scheme will utilizes the same images and texts in several tasks with different labels. This scheme is similar to a single-label CIL problem though with overlapped sample across tasks. A more reasonable setting is to learn all the labels of a sample once encountering it.
2. The identified problems are not clearly related to the continual learning or class-incremental learning. Althogh this paper specifies that semantic confusion and TN-FP confusion hinders the MLCIL, these problems are not directly related to CIL and they are likely the problems within multi-label learning. Do these two problems intensify the catastrophic forgetting or hinder the knowledge transfer between different tasks? Corresponding illustration should be included to elaborate the motivation of this study.
3. The ulitilization of PCD seems not practical. The PCD requires the total number of categories to perform a curriculum-based learning. However, a continual learning system may not identify the number of classes in the learning tasks.
4. It is strange to utilize CLIP as the base model. CLIP is a vision-language model and the continual learning based on CLIP has further issue tailored in multi-modal models. This paper does not present motivation about using CLIP, a VLM, to conduct the MLCIL . There are several previous works of MLCIL utilizing the TResNetM [1,2], and it can be better within the context of this paper.
5. Lack of results of dataset with large-scale categories. The DDP needs to store class-speciifc prompts and does not keep a selector. If there are a large scale of catergories, this manner may lead to knowledge confusion since all prompts are inserted. Also, large computational overhead will be introduced. Corresponding experiments should be included to verify this issue.
6. Lack of statistical results of experiments. The experiments are conducted once with only one order of classes. Experiments with more orders and multi-run results should be included to further verify the robustness of the proposed DDP.

To sum up, this paper should improve the presentation, illustration of the motivation, and experiments.

[1] Kaile Du, et al. Confidence Self-calibration for Multi-label Class-Incremental Learning. In ECCV 2024.

[2] Xiang Zhang, et al. L3A: Label-Augmented Analytic Adaptation for Multi-Label Class Incremental Learning. In ICML 2025.

**Questions:**

Please refer to the weaknesses.

---

### Official Review · Reviewer_U2NQ · 2025-10-28

**Soundness:** 2
**Presentation:** 3
**Contribution:** 2
**Rating:** 4
**Confidence:** 2

**Summary:**

The authors propose DDP, a replay-free, parameter-efficient framework: it uses class-specific positive-negative prompts to resolve semantic confusion, PCD to suppress false positives, freezes past prompts as knowledge anchors, and adopts interlayer prompting for efficiency. DDP outperforms prior methods on MS-COCO and PASCAL VOC, being the first replay-free MLCIL approach exceeding 80% mAP and 70% F1 on MS-COCO B40-C10.

**Strengths:**

- The paper makes a valuable contribution by targeting MLCIL’s unique challenges—semantic and TN–FP confusion—that prior prompt-based methods ignored.
- Its replay-free, parameter-efficient nature aligns with practical deployment needs, and the empirical study (two datasets, diverse baselines, ablations) demonstrates consistent improvements.

**Weaknesses:**

- Lack of comparison with existing very recent methods, such as [1] and [2], many of which have higher performance:

[1] L3A: Label-Augmented Analytic Adaptation for Multi-Label Class Incremental Learning

[2] Confidence self-calibration for multi-label class-incremental learning

- The paper makes key claims on the efficiency, yet no important results are present regarding this contribution. Although some results are in the appendix, there are no comparison with existing methods, only its vanilla counterpart.

**Questions:**

See weaknesses. My raiting will be adjusted according to the responses to all the reviewers' points.

---

### Meta-Review · Area_Chair_HD5Y · 2026-01-07

**Summary:**

In this paper, the authors propose Dual-Decoupled Prompting, a replay-free and parameter-efficient framework for class-incremental learning. The paper was reviewed by four expert reviewers, and most of them gave negative ratings. The only positive rating is a borderline acceptance from Reviewer 43au. The reviewers raised many concerns, such as a lack of comparison with existing very recent methods and a lack of implementation details on interlayer prompting. The authors decided not to provide responses during the rebuttal stage. Therefore, it is a clear rejection.

**Reviewer Concerns:**

The paper was reviewed by four expert reviewers, and most of them gave negative ratings. The only positive rating is a borderline acceptance from Reviewer 43au. The reviewers raised many concerns, such as a lack of comparison with existing very recent methods and a lack of implementation details on interlayer prompting. The authors decided not to provide responses during the rebuttal stage.

**Reviewer Scores:**

The paper was reviewed by four expert reviewers, and most of them gave negative ratings. The only positive rating is a borderline acceptance from Reviewer 43au. The reviewers raised many concerns, such as a lack of comparison with existing very recent methods and a lack of implementation details on interlayer prompting. The authors decided not to provide responses during the rebuttal stage.

---

### Decision · Program_Chairs · 2026-01-26

Reject